# Assessment of Primary Cell Wall Nanomechanical Properties in Internal Cells of Non-Fixed Maize Roots

**DOI:** 10.3390/plants8060172

**Published:** 2019-06-13

**Authors:** Liudmila Kozlova, Anna Petrova, Boris Ananchenko, Tatyana Gorshkova

**Affiliations:** 1Kazan Institute of Biochemistry and Biophysics, FRC Kazan Scientific Center of RAS, Lobachevsky Str. 2/31, Kazan 420111, Russia; anna.petrova@kibb.knc.ru (A.P.); gorshkova@kibb.knc.ru (T.G.); 2Nanotechnology Research and Education Center, Vyatka State University, Moskovskaya Str. 36, Kirov 610000, Russia; boris.f5112@gmail.com

**Keywords:** Atomic-force microscopy (AFM), nanomechanical properties, primary cell wall, root, *Zea mays*

## Abstract

The mechanical properties of cell walls play a vital role in plant development. Atomic-force microscopy (AFM) is widely used for characterization of these properties. However, only surface or isolated plant cells have been used for such investigations, at least as non-embedded samples. Theories that claim a restrictive role of a particular tissue in plant growth cannot be confirmed without direct measurement of the mechanical properties of internal tissue cell walls. Here we report an approach of assessing the nanomechanical properties of primary cell walls in the inner tissues of growing plant organs. The procedure does not include fixation, resin-embedding or drying of plant material. Vibratome-derived longitudinal and transverse sections of maize root were investigated by AFM in a liquid cell to track the changes of cell wall stiffness and elasticity accompanying elongation growth. Apparent Young’s modulus values and stiffness of stele periclinal cell walls in the elongation zone of maize root were lower than in the meristem, i.e., cell walls became more elastic and less resistant to an applied force during their elongation. The trend was confirmed using either a sharp or spherical probe. The availability of such a method may promote our understanding of individual tissue roles in the plant growth processes.

## 1. Introduction

All living organisms that have cell walls (plants, fungi, and bacteria) must deal with the need to find a balance between wall strength and extensibility, which is necessary for growth [1]. Regulation of the mechanical properties of the cell wall is a key parameter used by plants to control the growth behavior of individual cells and tissues [2]. There is a broad spectrum of experimental approaches that could be used to characterize the mechanical properties of cell walls [3]. However, atomic force microscopy (AFM) remains the primary method at the micro- and nanoscales. This technique uses a cantilever tip (probe) as a nanoindenter. The properties of the cell wall material are calculated from the penetration depth of the probe, which intrudes into the sample at a certain force [4]. The traditional objects for such studies are the outer tissues of *Arabidopsis* [5,6,7,8] and onion [9,10]. Some studies have been conducted on cell cultures of *Arabidopsis* [11] and other species, including Italian ryegrass [12], tomatoes [13], and wild type and transgenic rice [14].

Investigation of the cell walls of inner tissues of a plant is thought to be more or less impossible [4] because hand sectioning produces samples of a high degree of roughness that can lead to probe damage. One approach to overcome this obstacle is the preparation of sections from resin-embedded material and the use of diamond knives to obtain smooth surfaces [8,15]. The problem is that resins penetrate highly porous and multilayered cell walls and change their properties [16,17]. These changes are expressed at different extents for cell walls of different origin probably because of their porosity and lamellar structure [17]. Furthermore, these parameters, in turn, are known to vary among cell types and change during cell development [2], calling into doubt the reliability of any comparative studies using resin-embedded material.

The embedding procedure also includes fixation and dehydration steps. Both were shown to lead to substantial changes in the architecture and properties of living cells [18,19,20]. Dehydration also accompanies cryosectioning, making this approach similarly inapplicable to study nanomechanics of plant internal tissues, at least for cells with primary cell walls. Plant tissue also should not be dried during AFM measurements *per se*. Natural water evaporation from the sample surface may be enhanced by the laser beam, which is focused on the cantilever and inevitably raises the temperature in proximity to the specimen. Attempting to avoid sample dehydration is one of the main reasons why most previous research has been conducted in a liquid cell.

In order to conduct studies in a liquid cell, the biological sample has to be tightly bound to some substrate to avoid it being dragged by the tip. Investigations of plant tissues generally use two approaches: mounting of the samples in agarose [4] and fixation by glue [6]. Mechanical property measurements of intact cells in a liquid cell are confounded by the influence of turgor pressure [3]. Removal of the effects of turgor on mechanical properties is usually achieved through conducting experiments in osmolyte solutions [4,9,10]. However, different osmolytes have been shown to significantly change the mechanical properties and even the thickness of cell walls [9,19]. Plasmolytic conditions may also influence cell geometry, which in turn has considerable effects on measured values [21].

Here we present a method for AFM-based characterization of plant cell wall mechanical properties for internal tissues of plant organs using non-fixed and never-dried material of maize roots. Vibratome sectioning produced a surface smooth enough to be studied by an atomic force microscope and avoided drying during cutting. The use of the AFM liquid cell solved the problem of section drying during measurements. Because we investigated cell walls of the cut cells that had no turgor, we conducted our experiments in water. The stiffness and elasticity of stele vascular parenchyma periclinal cell walls in different regions of maize primary roots were characterized to test the hypothesis that a direct relationship exists between cell wall mechanical properties and the growth rate of cells. Investigations were conducted on both longitudinal and transverse sections to examine the anisotropy of the mechanical properties.

## 2. Results and Discussion

### 2.1. Plant Section Preparation to be Studied by AFM

We investigated both transverse (Figure 1a,b) and longitudinal (Figure 1c,d) sections of axial plant organs, derived using a vibratome. The vibratome uses a blade vibrating in a horizontal plane to cut the sample. The advantage is that, in contrast with hand sectioning, the vibratome produces slices so that the top and bottom sides are parallel to each other. The crucial issue that should be addressed at this stage is the sample orientation. In the case of axial organs, the section plane should be either perpendicular (for a transverse cut) or parallel (for a longitudinal cut) to the organ axis. Oblique sectioning results in cell walls that are tilted relative to the sample surface and cannot be probed for mechanical properties [3].

Soft samples are usually embedded in low melting point agarose to be cut by the vibratome. The proper orientation of an object within the agarose block was achieved during the two-step embedding (see Materials and Methods section for details). A plastic Petri dish was used as an AFM liquid cell. The additional layer of agarose on the bottom of the Petri dish retained the agarose block containing the root section. AFM was conducted in a contact mode in water. Using this approach, we obtained topography images for cross-sections of primary maize root and *Arabidopsis* hypocotyl (Figure 2) to demonstrate the applicability of the proposed method for studying specimens of different sizes. Vibratome sectioning produced a surface smooth enough to be studied by an atomic force microscope and prevented sample drying during cutting. Performing AFM in a liquid solved the problem of section drying during experiments.

### 2.2. Stele Tissue Topography and Force Spectroscopy

Two types of AFM probes were used in the study: a sharp conical probe with 10 nm apex radius, and a spherical probe with 100 nm sphere radius.

Two zones of maize primary roots were used for the experiments: the meristem zone (0–1 mm, measured from the root cap junction) and the beginning of the elongation zone (2–4 mm) (Figure 1d). Only one of these zones was studied on each longitudinal section to avoid prolonged incubation of plant material in water. Cells of vascular parenchyma in proximity to metaxylem strands were investigated (Figure 1b,d). Both the periphery and the central part of stele were avoided to exclude other tissues from the analysis. The average cell lengths in the meristem and elongation zones were 9.9 ± 1.8 μm and 40.0 ± 3.0 μm, respectively (Figure 3). These numbers were in agreement with cytological studies of these zones in the roots of maize [22,23]. The width of cells did not alter significantly between two zones and accounted for 12.1 ± 2.8 and 12.1 ± 3.8 μm, respectively. Both longitudinal and transverse sections of maize root were analyzed to examine the difference in cell wall biomechanical properties in two directions. Only periclinal cell walls were studied.

Figure 3 represents AFM topography images of maize roots obtained by spherical and sharp probes. No visible differences in XY-resolution were observed for these tips because of the low number of lines per μm. The height signal recording from both forward and backward passages of a cantilever allows for discrimination of the cell walls, which were not stable on a sample surface (shown by arrows). Such walls were not used for determination of the mechanical properties.

On longitudinal sections, both periclinal and anticlinal cell walls of vascular parenchyma and metaxylem were observed. On transverse sections, mostly periclinal cell walls were presented. Rarely the anticlinal cell walls of undamaged cells were visible (Figure 3e,f,g). These cells stayed uncut because they were below the section plane but relatively close to it. In bigger cells, like metaxylem, the curvature of these cell walls (Figure 3f) indicated the presence of turgor. On transverse sections, periclinal cell walls around undamaged anticlinal walls were not analyzed because of the possible influence of turgor on measured mechanical properties [3].

Topography imaging was performed in a contact mode with the constant force not exceeding 4 and 20 nN for CSG10 and B100, respectively. Force spectroscopy was carried out on periclinal cell walls in a point-to-point regime (dots in Figure 3) with a maximum indentation depth of 300 nm. Detected height variations of the sample surface were in a range of 5 to 7 μm. However, some lower parts of sectioned cells were beyond the Z-scanner capability and were registered by the microscope as zero height (Figure 2 and Figure 3).

Indentation rates used in different studies of plant cells varied between 0.5 μm s^−1^ [6] and 20 μm s^−1^ [21,24]. Depending on the loading rate, the elastic, plastic or viscous response of biomaterial may dominate [25]. Viscous behavior leads to a partial dissipation of deformation energy and appears as the lag between approaching and retraction curves. In this case, values of force are lower at the same displacement distance while the probe retracts than while it approaches. This phenomenon is known as hysteresis [26] (Figure 4a). The portion of deformation energy which was returned with the elastic response may be calculated as the ratio between the areas under retraction and approaching curves (Figure 4b). We did not observe substantial differences in hysteresis between curves recorded at 1, 2 and 5 μm s^−1^ (Figure 4b), and used 2 μm s^−1^ for all measurements.

The ideal force–displacement curve is presented in Figure 4a. Such curves had a non-tilted baseline, a clear contact point (the point on the approaching curve where the force gains a positive value), small hysteresis and adhesion (appears as negative force values on the retraction curve close to the contact point), and no shoulders or inhomogeneities on both the approach and retraction components. For the sharp probe, 45% of all recorded curves were good, while only 30% were good for the spherical tip. A sharp tip probably was more effective because of the lower ratio of tip radius to cell wall thickness. Attempts to get force–displacement curves from movable cell walls (white arrows in Figure 3) were much less resultative for both types of probe.

Artefact curves typical for sharp and spherical probes are shown in Figure 4c,d, respectively. Curves obtained by a sharp probe often had failures along the approaching part (Figure 4c). When studying viruses, protein shells and lipid bilayers, such loading curves usually were attributed to a rupture of a particle surface [27,28,29]. The sharp CSG10 tip resembles a needle in its geometry. It may pierce the cell wall.

Shoulders on the approaching curve, which represented typical artefacts for a spherical probe (Figure 4d), may be interpreted as a sign of sample buckling [30], or the tip slipping off a sample edge [31]. The artefacted approach curve was often paired by a standard retraction curve (Figure 4c,d). However, the latter cannot also be used for determination of the mechanical properties. Both approaching and retraction curves should be analyzed before Young’s modulus extraction. In a point-to-point regime, it may be carried out manually, while neuronal network implication was proposed for the force-mapping mode [31].

### 2.3. Stiffness and Elasticity of Maize Root Central Cylinder Cell Walls

Stiffness is the resistance of a solid body to deformation by an external force. It can be calculated directly by dividing the applied force by the deformation. A model of sample-indenter contact is not needed for such a calculation. This is an advantage because all available models of contact mechanics are based on assumptions that do not perfectly match the reality of living cell investigations. The reverse side of stiffness determination is that the result depends strongly on the geometry of the sample and indenter. The stiffness values that were obtained by different indenters should not be directly compared to each other [26].

The stiffness of vascular parenchyma cell walls in different zones of the maize root is presented in Table 1. The spherical indenter, in general, gives higher values of stiffness than the sharp one in agreement with the dependence of this property on the system geometry. The needle-like CSG10 tip meets less resistance pressing the cell wall than the spherical B100 tip with a 10-times larger radius. A. Zdunek and A. Kurenda [13] measured the stiffness of isolated tomato mesocarp cells by pyramidal and spherical tips with 20 nm and 10 μm radii of curvature, respectively. There was a three-fold difference in the stiffness values: lower indenter radius corresponded to a lower stiffness.

Stiffness values obtained for the meristem zone were confidently higher than those for the elongation zone of maize root for both sharp and spherical tips (Table 1). Thus, the transition of cells to elongation growth was accompanied by a decrease in the stiffness of the periclinal cell walls of maize root vascular parenchyma. Growing apical parts of *Papaver* [32] and *Lilium* [33] pollen tubes were characterized by lower apparent stiffness than distal parts of the same tubes. These experiments highlighted the heterogeneity of stiffness within one cell, where different parts of the cell had different growing capabilities. Both meristematic and elongating cells of maize root grew, but with different rates. Classical studies of maize root growth postulate the relative elongation growth rate is minimal in the first mm of the root and increases significantly up to the fourth mm [34,35]. Thus, higher stiffness corresponded to a lower growth rate and *vice versa* in maize root stele cells.

Elasticity is a property of a material to deform reversibly under external force (stress). For ideal elastic contacts, the force–deformation (stress–strain) relationship is linear. The Young’s modulus is the ratio of applied stress to observed strain, expressed in Pascals (Pa) [26]. The Young’s modulus is determined from the slope of the linear region of a force–indentation curve. There are several models of contact mechanics used to calculate it from AFM force–indentation curves. The Hertz model and its modification by Sneddon describe the spherical and conical probes correspondingly in contact with the flat surface. They can be used for soft materials if adhesion forces are low [25,26]. We did not observe substantial adhesion during the experiment and thus used the Hertz/Sneddon model.

The Hertz model requires a linear response of an isotropic and homogeneous sample and considers the probe a perfect sphere, which perpendicularly indents a non-corrugated plane surface [25]. All assumptions are not strictly valid for studying living cells by AFM [25,26]. To stress this point, the derived elastic moduli are called “apparent” Young’s moduli.

Mean values of apparent Young’s moduli between 1.5 and 6.7 MPa were found for periclinal cell walls of maize root vascular parenchyma (Table 1). Young’s modulus values in a similar range have been previously reported for *Lolium* cultured cells [12], onion epidermis [9,10], *Arabidopsis* hypocotyls [24], cotyledons [21], and shoot meristems [5].

Young’s moduli and their standard deviation obtained by a sharp probe were higher than those obtained by a spherical one (Table 1). Higher Young’s moduli measured by a sharp tip compared to a colloidal tip have been described before for *Arabidopsis* cotyledon epidermis [21] and isolated tomato mesocarp cells [13]. The difference in measured properties observed between sharp and relatively blunt probes is usually associated with the ability of the sharper tip to resolve the structures at a subcellular scale [13,21,26].

Either sharp or spherical probes gave significantly different apparent Young’s moduli in the meristem and at the beginning of the elongation zone of maize root (Table 1). In *Arabidopsis* shoot meristem, lower apparent Young’s moduli were detected on the flanks, where cells exhibited fast growth compared with the meristematic apex, where growth occurred much slower [5]. An increase in local cell wall elasticity (i.e., lower Young’s modulus) was shown to precede the primordia formation in *Arabidopsis* shoot meristems [36]. Similarly, elongating longitudinal (axial, periclinal) cell walls of *Arabidopsis* hypocotyl epidermal cells were found to be more elastic than transverse (anticlinal) cell walls, where growth was slower [24,37]. Boudon et al. [38] and Fayant et al. [39], using computational modelling, confirmed the necessity of an increase in local cell wall elasticity for primordia initiation and for polar growth of the pollen tube, respectively. Here, we have shown that the elongation process in maize root stele cells is accompanied by a decrease in stiffness and apparent Young’s modulus of periclinal cell walls.

No differences were observed between the results obtained on transverse and longitudinal sections, either in stiffness or in Young’s modulus (Table 1), suggesting the absence of anisotropy within one zone of maize root. However, the anisotropy may be hidden by natural variations of the measured parameters. Another possible explanation is that the anisotropy of the mechanical properties in the investigated zones had not developed yet to an extent that could be detected.

### 2.4. Potential Limitations of the Proposed Method

Water is not a physiological solution for plant tissues. Inevitable leaching of ions and small molecules occurs during both sectioning and scanning. For example, calcium and boron are known to play an essential role in the mechanical performance of plant cell walls due to their ability to crosslink pectins [10,24,40], however, this mechanism may be not crucial for cereal cell walls having low pectin content [41]. Wounding while cutting generates a reactive oxygen species burst, which also may affect the cell wall biochemistry [42] and, hence, the properties. Simultaneously, autolytic enzymes within cell walls may still be operational in cut cells, while synthesis and deposition of new portions of polysaccharides are doubtful. All of these issues could not be addressed simultaneously without bringing other errors into the AFM measurements. Thus, the time of the experiment should be standardized and shortened as much as possible. In the current study, each root of maize was used to produce only one section which was used for the analysis of only one zone. All procedures starting from root excision to last force-displacement curve recording were performed within one hour.

During AFM, cut cell walls are incubated in water for at least half an hour. Some changes in cell wall mechanical properties may occur within this time. Each graph in Figure 5 shows apparent Young’s moduli measured successively within 30 minutes at different points on one root section. Thus, the set of measurements for each variant in Figure 5 is taken from a single representative root. Although the substantial variation in the obtained values was observed, no clear changing trends in Young’s moduli over time were established (Figure 5). This suggests that incubation of cut cell walls in water during AFM did not lead to detectable one-directional changes in cell wall elasticity.

Some alterations in cell wall properties may also have occurred before measurements, such as in the vibratome bath while sectioning. Vibratome sections are widely used for immunodetection of cell wall epitopes [43,44], indicating the persistence of cell wall components on the surface of the section during its preparation. The wounding itself causes a rearrangement of the cell wall structure and mechanics; however, this response requires several hours to become pronounced [45,46]. The tissue reaction on cutting by a vibratome may be even slower because of the dissipation of signal molecules in the vibratome bath where the movements of the blade constantly mix water. Nevertheless, the material cut using a vibratome cannot be considered as alive. However, in short term investigations, the mechanical properties measured on such a material may be closer to an in vivo state than those obtained as a result of any other sample preparation procedure.

## 3. Materials and Methods 

### 3.1. Plant Material and Sample Preparation

*Arabidopsis* (*Arabidopsis thaliana*, cv. Columbia) seedlings were grown on a wet paper for 3 days in the dark at 27°C. The upper part of the hypocotyl was used for AFM topography imaging. Maize (*Zea mays*, cv. Interkras 375) seedlings were grown hydroponically for 4 days in the dark at 27°C. Apical 6 mm segments of primary root were used for sectioning (Figure 1c). Cells of stele vascular parenchyma in the meristem zone (0–1 mm of the root apex, measured from the root cap junction) and in the first half of the elongation zone (2–4 mm of the root apex, measured from the root cap junction) were analyzed. The distribution of growth zones in maize primary roots has been described and proven [22]. For the dissection by the vibratome, the excised root apex was embedded in 3% (*w*/*w*) low melting point agarose.

Melted agarose was poured in a Petri dish to form a layer with a horizontal surface. After it cooled and partially solidified, the segment of a root (or a hypocotyl) was placed on its surface and then covered by another layer of agarose. Thus, a strictly horizontal position of the organ was achieved. After complete congealing of agarose, the block, which contained the root segment, was cut out using a razor blade. The block was mounted on a vibratome stage vertically or horizontally depending on what kind of section was prepared using cyanoacrylate adhesive (Best-CA, Germany). A Leica VT 1000S (Leica Biosystems, Germany) vibratome (blade speed—1.3 mm s^−1^, blade frequency—90 Hz) was used to obtain 400 µm thick sections of plant material. We used sections of 400 µm thickness because they are easier to handle. However, the sample thickness could be increased or decreased depending on the object’s dimensions or the research goal. The obtained sections were caught from the vibratome bath on a glass slide using a brush.

Melted agarose was poured on the bottom of another Petri dish, which was used then as an AFM liquid cell. Before complete congealing of this agarose layer, the section still surrounded by agarose from the initial block was moved from the glass slide to the agarose layer in the Petri dish (Figure 1a,c). The sample was then covered with water to be studied by AFM.

### 3.2. Atomic Force Microscopy

AFM investigations were performed at room temperature in a liquid cell using a NTEGRA Prima (NT-MDT, Russia) microscope. Topography images were obtained in a contact mode using CSG10 AFM tips (NT-MDT, Russia) with a typical resonant frequency of 22 kHz, average spring constant of 0.11 N m^−1^ and apex radius of 10 nm, and spherical Biosphere-B100-CONT (NanoTools, Germany) tips with a sphere curvature radius of 100 nm and average spring constant of 0.2 N m^−1^. The thermal tune procedure was performed for each new cantilever to determine its unique spring constant. Deflection sensitivity was determined at room temperature in water on a fresh cover glass for each new cantilever, between samples, and every time after laser adjustment. Scanning was conducted at a speed of 5 s per line. The typical scanning area was 50 × 50 μm with 64 × 64 point resolution. Scanning was carried out in a regime of controlled force with a maximum force acting on the sample of 4 and 20 nN for CSG10 and B100, respectively. The AFM images had no correction for noise or surface subtraction.

Force–displacement curves were recorded at different points of the periclinal cell walls after the topography images were obtained. The indentations were performed at a rate of 2 μm s^−1^. Sample indentation did not exceed 300 nm for both types of probe, applied force did not exceed 20 nN for the sharp CSG10 tip and 100 nN for the spherical B-100 tip. Obtained force–displacement curves were analyzed on the presence of artefacts (Figure 4). Selected curves were used to determine Young’s moduli and cell wall stiffness.

### 3.3. Calculations

Apparent Young’s moduli were calculated from the retraction part of the force–displacement curves using FCProcessor2 script of Nova Px 3.4 software (NT-MDT, Russia) with the following input parameters: deflection sensitivity [m A^−1^]; cantilever stiffness [N m^−1^]; tip radius [nm]; half-angle of a cone [°]; contact mechanic model to fit [Hertz sphere or Hertz cone, i.e., Sneddon]. Poisson’s ratio was 0.5. The FCProcessor2 script determines the baseline and converts the force–displacement curve into a force–deformation curve based on deflection sensitivity and cantilever stiffness. It then conducts the fitting to the chosen model using data on tip geometry. The region of the curve between 20% and 80% of the maximum applied force was used for fitting. Output parameters for each analyzed force–displacement curve were deformation [nm] and Young’s modulus [MPa]. For calculation of stiffness [N m^−1^], maximum applied force was extracted as the maximum point of the force–displacement curve and divided by the deformation obtained from the curve analysis. Areas under approach and retraction curves for hysteresis evaluation were calculated by Simpson’s method.

### 3.4. Statistics

Once it was cut, the root was dissected transversely or longitudinally and used for biomechanical characterization of either the meristem or elongation zone. Ten roots were examined for their mechanical properties for each zone (meristem and elongation zone), each type of the section (transverse and longitudinal), and each type of probe (sharp and spherical tips), giving 80 roots in total. Between 20 and 50 force–displacement curves were recorded on each section. The mean value of apparent Young’s moduli and stiffness from one section was used as one biological replica. Mean values with standard deviations among biological replicates are presented. Mean separation was performed by ANOVA followed by Tukey test at α = 0.01, using the SPSS software package (v.21, IBM Corp.).

## 4. Conclusions

A properly oriented piece of plant material can be sectioned by a vibratome to produce a surface appropriate for AFM, formed by cut cell walls of inner tissues. This approach is applicable to specimens of different sizes. Force–displacement curves can be recorded in different points of the scanned area to calculate stiffness and the apparent Young’s modulus. Some force–displacement curves had artefact character that might occur due to cell wall bending or the probe slipping off the sample surface. Such curves can be distinguished by their shape and should be excluded from the analysis. The sharp tip used in the current study was characterized by a higher proportion of non-artefact curves compared to the spherical one. At the same time, the standard deviations of measured parameters were always higher for the sharp tip, possibly because of its ability to resolve the substructures of cell walls. Independently of the probe type, meristem cell walls of maize root were characterized by higher stiffness and Young’s modulus values compared to walls at the beginning of elongation, confirming the idea of a direct relationship between growth rate and cell wall mechanical properties. Contemporary theories that claim a restricting role of a particular tissue in plant growth cannot be confirmed without direct measurement of the mechanical properties of internal tissue cell walls. The availability of such a method may promote our understanding of the role of individual tissues in plant growth performance.

## Figures and Tables

**Figure 1 plants-08-00172-f001:**
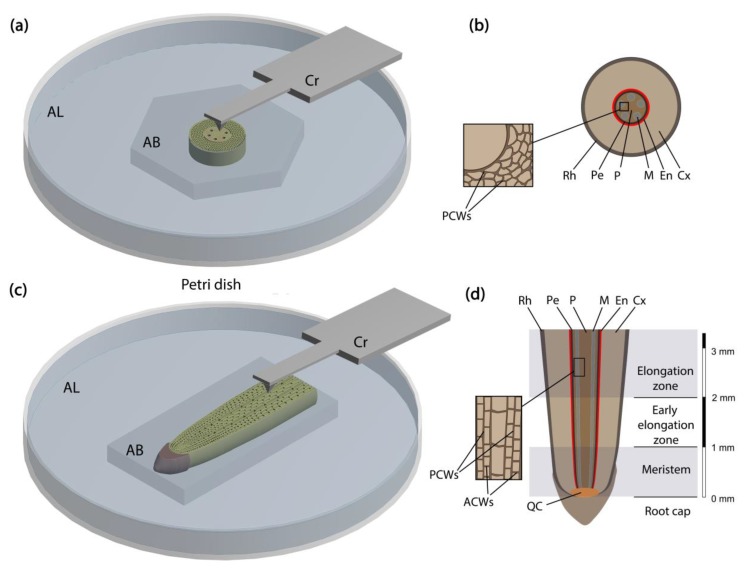
Scheme of sample preparation. General view of transverse (**a**) and longitudinal (**c**) sections of maize root mounted in a plastic Petri dish for atomic force microscopy (AFM). Tissue structure of maize root on transverse (**b**) and longitudinal (**d**) section with the inset showing the location of vascular parenchyma. Root zones distribution according to Kozlova et al. [22], grey semitransparent contours show regions which were investigated (**d**). AB—agarose block, AL—agarose layer, Cr—cantilever, Rh—rhizodermis, Cx—cortex, Pe—pericycle, P—pith, M—metaxylem, En—endodermis, PCW—periclinal cell wall, ACW—anticlinal cell wall, QC—quiescent centre.

**Figure 2 plants-08-00172-f002:**
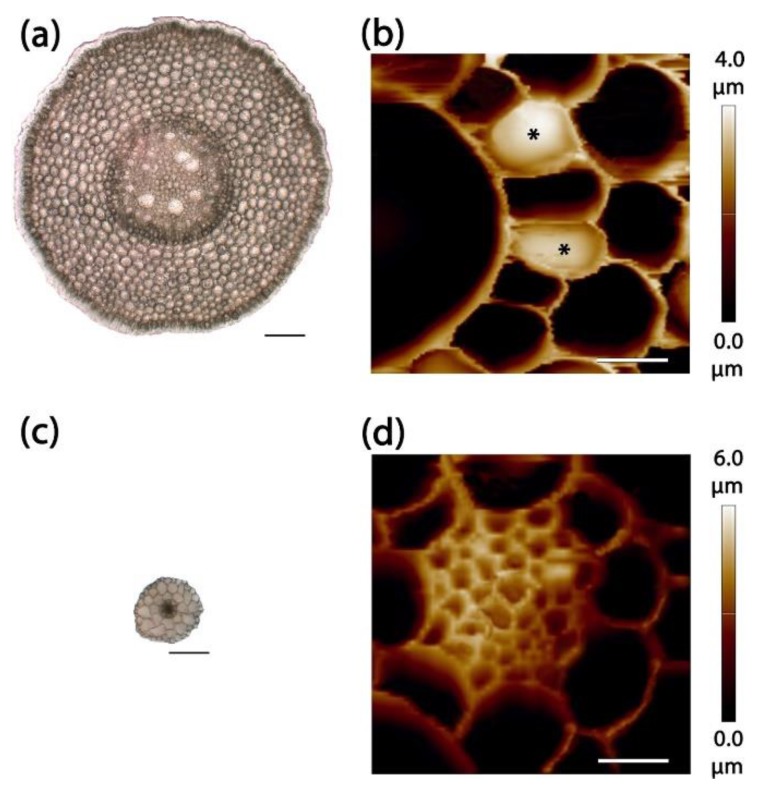
Transverse sections of primary maize root (**a,b**) and *Arabidopsis* hypocotyl (**c,d**). (**a,c**) light microscopy images; (**b,d**) AFM topography images. Bar (**a,c**) 100 μm; (**b,d**) 10 μm. Z-scale for (**b**,**d**) is presented on the right side of the corresponding image. Asterisks in (**b**) mark undamaged anticlinal cell walls of stele cells.

**Figure 3 plants-08-00172-f003:**
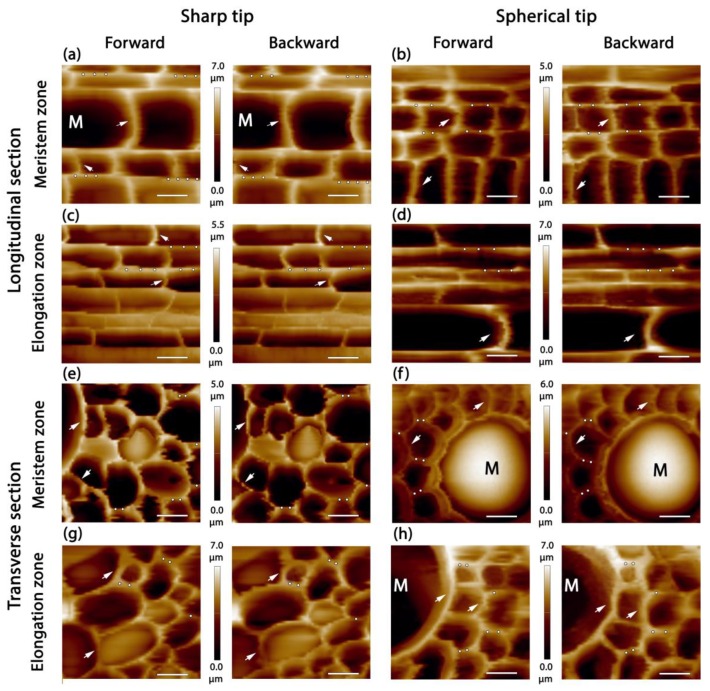
The topography of central cylinder cells imaged by sharp (**a**,**c**,**e**,**g**) and spherical (**b**,**d**,**f**,**h**) AFM tips on longitudinal (**a**–**d**) and transverse (**e**–**h**) sections of maize seedling primary root in the meristem (**a**,**b**,**e**,**f**) and elongation (**c**,**d**,**g**,**h**) zones. Each pair of images was generated by forward and backward height signals with 64 × 64 point resolution. M—metaxylem. White arrows show the displacement of some cell walls occurring at scanning. Dots represent points where force–displacement curves may be recorded. The bar for XY-dimension—10 μm. Z-scale is given between each pair of images.

**Figure 4 plants-08-00172-f004:**
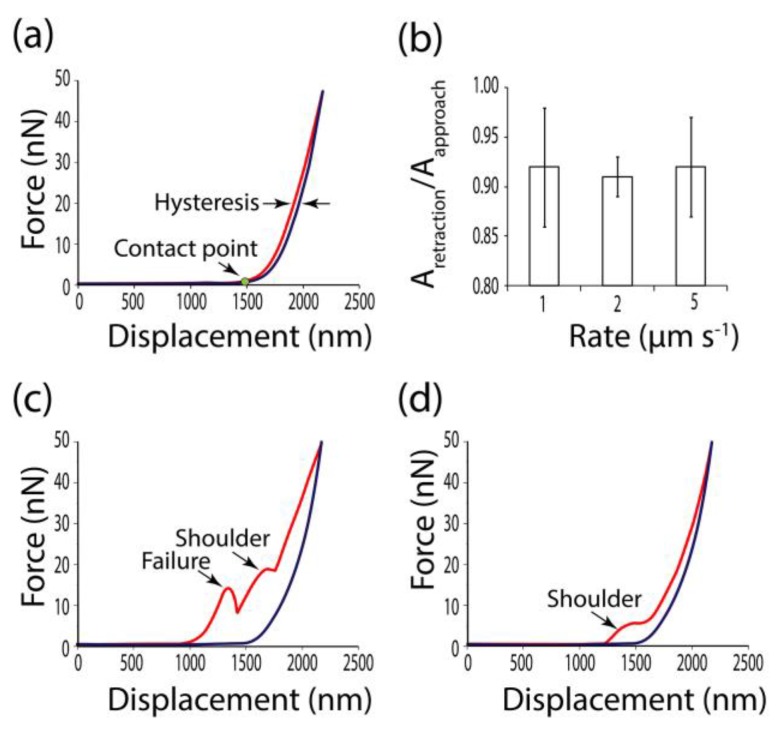
Force–displacement curves. Approach curves are red, and retraction curves are blue. (**a**) Normal curves, appropriate for calculations of apparent Young’s modulus. (**b**) Hysteresis observed at different loading rates expressed as the ratio of areas under the retraction and approach curves. Typical artefact curves for sharp (**c**) and spherical (**d**) probes.

**Figure 5 plants-08-00172-f005:**
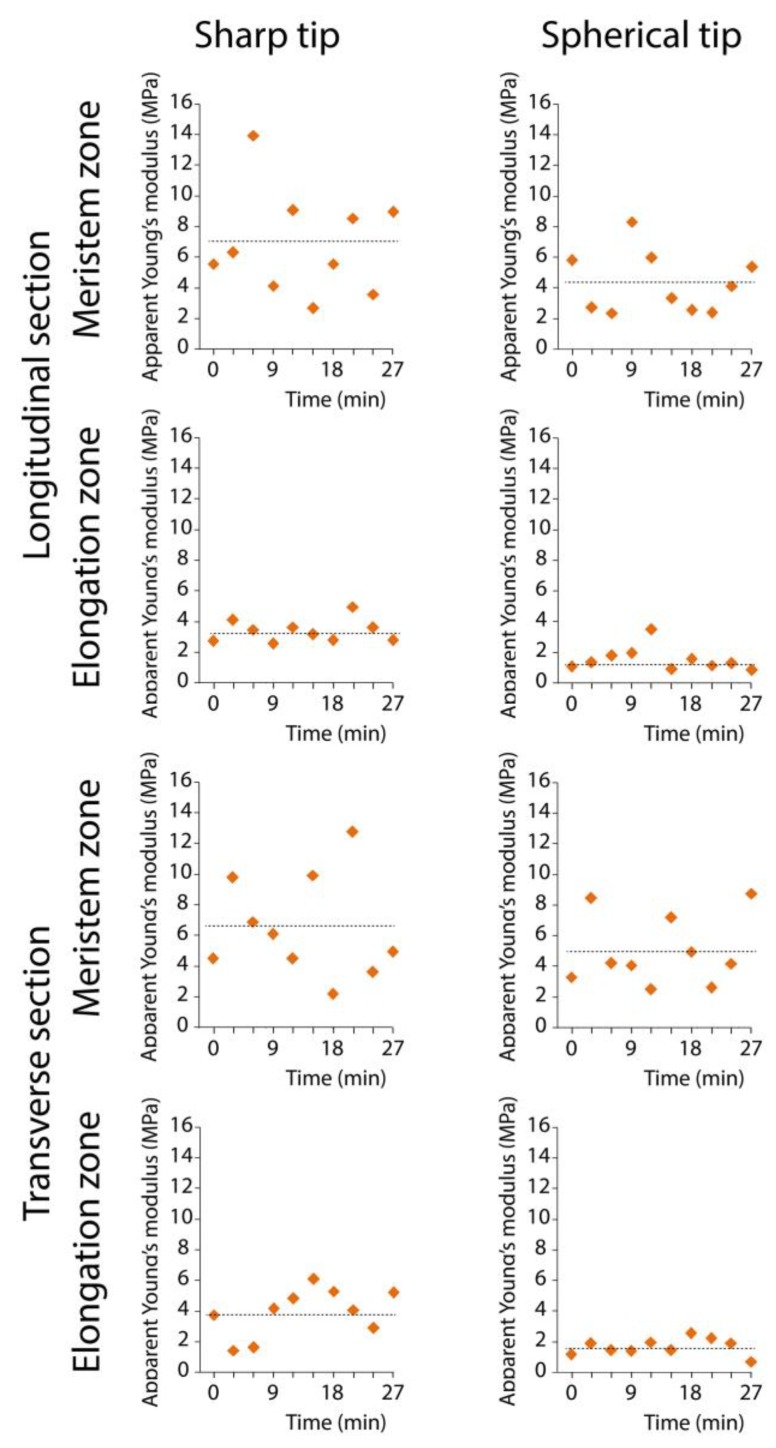
Changes of cell wall apparent Young’s moduli over time. The set of measurements for each variant is taken from a single representative root (i.e., one biological replica for each variant is shown). Ten markers represent values of apparent Young’s modulus obtained at different points on one maize root section within 30 minutes. The horizontal dotted lines correspond to mean value for this biological replica.

**Table 1 plants-08-00172-t001:** Mechanical properties of maize root periclinal cell walls. Mean values for biological replicates with standard deviations are presented. Different letters above the meanings within one row correspond to a significant difference according to one-way ANOVA followed by Tukey test at α = 0.01. Ten roots were examined for their mechanical properties for each zone (meristem and elongation zone), each type of the section (transverse and longitudinal), and each type of probe (sharp and spherical tips), total n = 80.

		Meristem Zone, Longitudinal Section	Elongation Zone, Longitudinal Section	Meristem Zone, Transverse Section	Elongation Zone, Transverse Section
Apparent stiffness (N m^−1^)	**CSG10**	0.07±0.01^a^	0.05±0.01^b^	0.07±0.01^a^	0.05±0.01^b^
	**B100**	0.56±0.13^a^	0.27±0.04^b^	0.58±0.07^a^	0.25±0.04^b^
Apparent Young’s modulus (MPa)	**CSG10**	6.46±1.02^a^	3.75±0.43^b^	6.73±1.59^a^	3.47±0.51^b^
	**B100**	4.26±0.97^a^	1.65±0.34^b^	4.54±0.83^a^	1.44±0.29^b^

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
