# Peer review of "Assessment of Primary Cell Wall Nanomechanical Properties in Internal Cells of Non-Fixed Maize Roots"

_plants, 2019, doi:10.3390/plants8060172_

Round 1

Reviewer 1 Report

General Comment:

 The manuscript by Kozlova et al. is focused in the cella wall properties at nanomolecular scale. The authors used the Atomic-force microscopy (AFM) for obtain the awesome view of the cell wall. In my opinion the biological significance is great, and this is a interesting paper, will be good contribution for kind of article in plants.

Particular comments:

Introduction: Line 54:  extend "AFM". 

(some in the abstract) extend "AFM".

Author Response

Thank you for comments.

Corrections were made accordingly.

Introduction: Line 54: extend "AFM".

(some in the abstract) extend "AFM".

corrected

Reviewer 2 Report

@page { margin: 0.79in } p { margin-bottom: 0.1in; line-height: 115% } a:link { so-language: zxx }

The authors present an new approach for mechanical characterization of plants internal cell walls using Atomic force microscopy, while using no fixation or embedding. This is interesting because it may allow a mechanical characterization of such internal cell walls without the bias induced by fixation/embedding. The authors also make a much appreciated effort at pointing out the possible limitations of their technique throughout the manuscript. However I have some concerns about the nature of the samples analyzed and thus the main conclusions of the paper stating that they could reveal a difference in mechanical properties between meristematic and elongating tissues.

My major comments are:

- Although I am not an expert in maize roots anatomy, I am a little confused about the representations made in figure 1 in relation with the samples shown in figure 2 and 3. The proportions and distribution of different tissue types as represented in figure1 b and c does not seem to match the anatomy of the tissue in figure 2a. Also in figure 1b and c there is no representation of the actual diversity of cell types in the stele (Endodermis, pericycle, phloem, xylem, protoxylem, sclerenchyma and pith), which can be problematic because these different cell types may have very different cell wall mechanical properties. Throughout the study it is not clear which cell type is actually being assessed. It is only referred to as central stele, which I would interpret as the pith cells. However, while it is more difficult to identify on the longitudinal sections, in the images of transverse section in figure 3 i-l and o-p, the presence of the metaxylem indicates that the surrounding cells being measured are most likely from the sclerenchyma, while the appearance of the cells in figure 3 m-n are of pith cells (as revealed by the tricellular junction spaces). In the frame of this analysis they should not be comparable as they are expected to have very different cell wall mechanical properties. If more pith cells than sclerenchyma (or phloem, xylem, protoxylem,...) cells are being analyzed in one case or another this should create a bias. The authors should clarify which cell types are actually being analyzed and if possible provide an additional file/figure showing the AFM topography scans and points where force displacement curves were measured on each scan (like in figure 3 but extended to the 80 biological replicates) to allow the reader to have a clear understanding of what is being measured and what type of measurement is behind the quantification made in table1. Without this, the validity of the main conclusion of the paper correlating growth and mechanical properties remains difficult to assess. If the authors cannot provide such additional figure showing measurement of homogeneous cell type for each case for actual comparison (e.g. only pith cells) I believe the authors should either perform additional measurement to provide appropriate data, or clearly tone down the biological conclusion and point to the bias induced by their measurement of heterogeneous cell types with various mechanical properties.

- The potential bias that I describe above could actually be used as an internal control to further validate the methods of mechanical quantification without embedding, using the existing data. For instance if the authors can correlate and differentiate the stiffness of different cell wall based on their cell type, ideally from a single scan encompassing different cell types (although this might be difficult given the size of the scan area...). Sclerenchyma cells’ cell walls should be much stiffer than the pith’s.

- In table 1 it should be made clear that there are ten biological replicate per measurement shown, rather than simply stating “n=80”. Also the authors could describe in the legend how many force displacement curves it actually represents (20-50 per biological replicate if I understood well from the method section). To me it was also a bit unclear whether the data represented in figure 5 were actually the same a those summarized in table1. It took me some time to understand that it was not the case so I think the description of the samples and measurements presented should be made clearer in general.

Additional minor comments:

- Fig1, The lines drawn pointing to the periclinal and anticlinal cell wall are not clear. Maybe a close up around a few cell would help to clearly define for the readers which are the periclinal and anticlinal cell walls.

- Line 112, the authors state that there are no pectins in cereal cell wall (“cereal cell walls lacking pectins”). While there are indeed lower content of pectin such cell walls, they are not absent. This should be corrected.

- Line 131, it is not clear if elongation rate was measured in this study and how, or if it is from the previously cited paper. Please clarify.

- Line 189, Section title should start with “2.3” instead of “2.2”.

- Line 221, 224 and 329, I believe the authors meant “Hertz” instead of “Herz”

- Fig3, The legend should be made more clear in the cases pointing out sharp vs conical tips and forward vs backward scan, notably by naming the panels rather than saying “two left columns” or “on the left”. Such indications (conical tip, spherical tip, backward scan, forward scan) could also be written directly on the figure (e.g. at the top of the image columns as well as at the left of the image rows for longitudinal, transverse, meristem and elongation) to make the figure more readily readable.

Author Response

Thank you for your careful analysis of our work and your valuable comments. Our answers are arranged in the table and presented below.

Although I am not an expert in maize roots anatomy,   I am a little confused about the representations made in figure 1 in relation   with the samples shown in figure 2 and 3. The proportions and distribution of   different tissue types as represented in figure1 b and c does not seem to   match the anatomy of the tissue in figure 2a. Also in figure 1b and c there   is no representation of the actual diversity of cell types in the stele   (Endodermis, pericycle, phloem, xylem, protoxylem, sclerenchyma and pith),   which can be problematic because these different cell types may have very   different cell wall mechanical properties.

Figure 1 has been redrawn to match the reality of maize anatomy. However, it is still a schematic view. Endodermis, pericycle and pith are shown, while phloem, protophloem, protoxylem and sclerenchyma are not. Partially to avoid the redundancy and partially because some tissues   (like sclerenchyma or protoxylem) are not differentiated yet while cells are at the stage of most active elongation (Fahn, 1990).

Throughout the study it is not clear which cell type   is actually being assessed. It is only referred to as central stele, which I   would interpret as the pith cells. However, while it is more difficult to   identify on the longitudinal sections, in the images of transverse section in   figure 3 i-l and o-p, the presence of the metaxylem indicates that the   surrounding cells being measured are most likely from the sclerenchyma, while   the appearance of the cells in figure 3 m-n are of pith cells (as revealed by   the tricellular junction spaces). In the frame of this analysis they should   not be comparable as they are expected to have very different cell wall   mechanical properties. If more pith cells than sclerenchyma (or phloem,   xylem, protoxylem,...) cells are being analyzed in one case or another this   should create a bias. The authors should clarify which cell types are   actually being analyzed and if possible provide an additional file/figure   showing the AFM topography scans and points where force displacement curves   were measured on each scan (like in figure 3 but extended to the 80   biological replicates) to allow the reader to have a clear understanding of   what is being measured and what type of measurement is behind the   quantification made in table1. Without this, the validity of the main   conclusion of the paper correlating growth and mechanical properties remains   difficult to assess. If the authors cannot provide such additional figure   showing measurement of homogeneous cell type for each case for actual comparison   (e.g. only pith cells) I believe the authors should either perform additional   measurement to provide appropriate data, or clearly tone down the biological   conclusion and point to the bias induced by their measurement of   heterogeneous cell types with various mechanical properties.

Thank you for noting this! Images which were   presented in Figure 3m, n and g, h indeed belonged to the pith, while the main volume of data presented in the article was obtained on cells of stele parenchyma in proximity to metaxylem strands. According to Esau (1965), this tissue can be referred to as vascular parenchyma. Indication of investigated tissue type was included in the article. Figure 1 was supplemented by the insets showing the appearance of studied cells on both longitudinal and transverse sections. Later in their development, these cells will become sclerified while pith cells will not (Esau, 1965). We have replaced images in   Figure 3m, n and g, h and excluded the force-displacement curves obtained on pith cells from the dataset. Then we have performed additional measurements to bring the number of biological replicates back to 10 for each zone, each type of probe and each type of section. Consequently, the numbers in Table 1   were slightly changed, however, insignificantly.

The potential bias that I describe above could   actually be used as an internal control to further validate the methods of   mechanical quantification without embedding, using the existing data. For   instance if the authors can correlate and differentiate the stiffness of   different cell wall based on their cell type, ideally from a single scan   encompassing different cell types (although this might be difficult given the   size of the scan area...). Sclerenchyma cells’ cell walls should be much stiffer   than the pith’s.

The comparison of different tissues and cells on advanced stages of development are planned for future work. The present article is aimed at the method introduction.

In table 1 it should be made clear that there are   ten biological replicate per measurement shown, rather than simply stating   “n=80”. Also the authors could describe in the legend how many force   displacement curves it actually represents (20-50 per biological replicate if   I understood well from the method section).

The required information was added.

To me it was also a bit unclear whether the data   represented in figure 5 were actually the same a those summarized in table1.   It took me some time to understand that it was not the case so I think the   description of the samples and measurements presented should be made clearer   in general.

The legend for Figure 5 has been changed

Fig1, The lines drawn pointing to the periclinal and   anticlinal cell wall are not clear. Maybe a close up around a few cell would   help to clearly define for the readers which are the periclinal and   anticlinal cell walls.

Figure 1 has been corrected accordingly

Line 112, the authors state that there are no   pectins in cereal cell wall (“cereal cell walls lacking pectins”). While   there are indeed lower content of pectin such cell walls, they are not   absent. This   should be corrected.

corrected

Line 131, it is not clear if elongation rate was   measured in this study and how, or if it is from the previously cited paper. Please clarify.

We decided to delete these numbers from present work and use references instead.

Line 189, Section title should start with “2.3”   instead of “2.2”.

corrected

Line 221, 224 and 329, I believe the authors meant   “Hertz” instead of “Herz”

corrected

Fig3, The legend should be made more clear in the   cases pointing out sharp vs conical tips and forward vs backward scan,   notably by naming the panels rather than saying “two left columns” or “on the   left”. Such indications (conical tip, spherical tip, backward scan, forward   scan) could also be written directly on the figure (e.g. at the top of the   image columns as well as at the left of the image rows for longitudinal,   transverse, meristem and elongation) to make the figure more readily   readable.

Figure 3 has been changed. Indications for the shape of the tip, section type, scanning regime and zone investigated are now given in the Figure.

Figure 5 is changed similarly

Reviewer 3 Report

Line 3: replace root with roots

Line 15: replace restricting with restrictive

Line 25: processes

Line 26: replace nanomechanichal with nanomechanical

Line 58: add a comma after liquid cell

Line 65: replace has with have

Line 69: replace using with use

Line 80: The vibratome uses a blade vibrating in a horizontal plane to cut the sample. – This should be a part of Materials and Methods

Line 118: states ‘In the current study, each root of maize was used for the analysis of only one zone’ and line 126 states ‘Two zones of maize primary roots were used for the experiments’. This ambiguity needs to be clarified

Line 167: Such curves had small hysteresis and adhesion, clear contact point, flat baseline, no shoulders or inhomogeneities on both approach and retraction components. These terminologies need to be defined to make it more comprehensible for a wider audience.

Line 341: each zone (mention those zones), each type of section (mention those sections)

1.    How is this technique scalable for different organs and tissues of the plants as the authors have cited that cell walls of different origin have different porosity and lamellar structure and the changes during cell development. Would the contact mode be applicable to all types of tissues especially for the soft tissues? Also, please consider discussing increasing the scalability for multiple samples.

2.    Addition of contrasting phenotypes to really validate the application of the technique would make it more widely acceptable. In Arabidopsis, a lot of T-DNA mutants are available that can be easily deployed. It is highly recommended to include at least two different genotypes to rule out the genetic component.

3.    Are the values in figure 5, for each section, correspond to the average of eight roots for each of the ten points? (Line 233 and 254)

4.    The manuscript has primarily focused on maize roots; however, the materials and methods also mention about Arabidopsis hypocotyls. The results of which are not explicitly stated. Please consider including them equally.

Author Response

Thank you for comments and advice. Our answers are arranged in the table and presented below.

Line 3: replace root with roots

Line 15: replace restricting with restrictive

Line 25: processes

Line 26: replace nanomechanichal with nanomechanical

Line 58: add a comma after liquid cell

Line 65: replace has with have

Line 69: replace using with use

corrected

Line 80: The vibratome uses a blade vibrating in a   horizontal plane to cut the sample. – This should be a part of Materials and   Methods

The parallelism of two sides of a produced section is one of the most important conditions which should be maintained. And we have to explain where it comes from. This is why we believe this point is relevant in its present place.

Line 118: states ‘In the current study, each root of   maize was used for the analysis of only one zone’ and line 126 states ‘Two   zones of maize primary roots were used for the experiments’. This ambiguity   needs to be clarified

Line 118 was extended to “In the current study, each   root of maize was used for the producing of only one section which was used   for the analysis of only one zone.” This paragraph is relocated now. It is placed at the end of the manuscript together with other possible limitations of the method.

“Only one of these zones was studied on each   longitudinal section to avoid prolonged incubation of plant material in   water.” Was added after Line 126

Line 167: Such curves had small hysteresis and   adhesion, clear contact point, flat baseline, no shoulders or inhomogeneities   on both approach and retraction components. These terminologies need to be   defined to make it more comprehensible for a wider audience.

The term “hysteresis” is now described in the paragraph above this phrase. Explanations for “contact point” and “adhesion”   are given in parenthesis. “Flat baseline” is changed on “non-tilted baseline”. Some corrections are made in Figure 4 as well.

Line 341: each zone (mention those zones), each type   of section (mention those sections)

corrected

How is this technique scalable for different organs   and tissues of the plants as the authors have cited that cell walls of   different origin have different porosity and lamellar structure and the   changes during cell development.

This technique is applicable for primary cell walls of different origin as was shown by the obtaining of the topography images of maize and Arabidopsis (see Figure   2). The method also allows the discrimination of cells belonging to one tissue at different stages of their development by their Young’s modulus and stiffness. The ability of the method to resolve the difference between tissues remains to be studied in future work.

Would the contact mode be applicable to all types of   tissues especially for the soft tissues?

The contact mode was successfully applied to animal cells, cell membranes and polymers which are much softer then plant cells (for review see Fufrêne et al., 2017). The most important parameter that should be taken into account here is the cantilever stiffness. It should be comparable with the stiffness of the investigated material (Krieg et al.,   2019). For the topography imaging of plant tissues with primary cell walls the cantilevers with the stiffness of 45 N/m producing the forces up to 1000   nN were applied (Braybrook, 2015). We used cantilevers three orders of magnitude softer than this, and correspondingly lower forces.

Also, please consider discussing increasing the scalability   for multiple samples.

Existing data do not allow us to discuss the scalability for multiple samples. Several roots or hypocotyls (or other plant organs) can be mounted in one agarose block and then sectioned to be studied by AFM. However, the multiplication will extend the time for both preparation and measurements. We maintained these times as low as possible. Additional investigations of tissue behaviour, while it is waiting for its turn to be studied, are necessary.

Addition of contrasting phenotypes to really   validate the application of the technique would make it more widely   acceptable. In Arabidopsis, a lot of T-DNA mutants are available that can be   easily deployed. It is highly recommended to include at least two different   genotypes to rule out the genetic component.

The capability of AFM to resolve the mechanical differences between wild-type plants and mutants was demonstrated several times (Peaucelle et al., 2015; Majda et al., 2017; Bou Daher et al., 2018).   For maize which serves as the main object of our scientific interest, the mutants are not that widespread and easily obtainable. The present article is   aimed at the method introduction. Future work may include the mutants as well. Thank you for advice)

Are the values in figure 5, for each section,   correspond to the average of eight roots for each of the ten points? (Line   233 and   254)

No, they don’t. Figure legend and corresponding text have been changed for clarification.

The manuscript has primarily focused on maize roots;   however, the materials and methods also mention about Arabidopsis hypocotyls.   The results of which are not explicitly stated. Please consider including   them equally.

Arabidopsis hypocotyls were used to show the applicability of our approach to the specimens of different sizes. For this, imaging is the most important part. Successful topography imaging enables further recording of force-displacement curves and their analysis which we didn’t perform. There are many groups studying Arabidopsis mechanics. We hope, they will find our method promising and adopt it.

Round 2

Reviewer 2 Report

All my points have been addressed and I think the paper is now good for publication.

Very interesting new approach for AFM!

Author Response

Thank you!